# Exploring bioactive compound origins: Profiling gene cluster signatures related to biosynthesis in microbiomes of Sof Umer Cave, Ethiopia

**Abu Feyisa Meka** [ID][1,2], **Gessesse Kebede Bekele**[1,3], **Musin Kelel Abas**[1,3], **Mesfin Tafesse Gemeda**[1,3]*

**1** Department of Biotechnology, Addis Ababa Science and Technology University, Addis Ababa, Ethiopia, **2** Department of Biology, Bule Hora University, Bule Hora, Ethiopia, **3** Biotechnology and Bioprocess Centre of Excellence, Addis Ababa Science and Technology University, Addis Ababa, Ethiopia

* mesfin.tafesse@aastu.edu.et

**Data availability statement:** Sequence data that support the findings of this study have been submitted to the NCBI Sequence Read Archive (SRA) under accession number PRJNA1082540, (https://www.ncbi.nlm.nih.gov/sra/PRJNA1082540) and Supporting

## Abstract

Sof Umer Cave is an unexplored extreme environment that hosts novel microbes and potential genetic resources. Microbiomes from caves have been genetically adapted to produce various bioactive metabolites, allowing them to survive and tolerate harsh conditions. However, the biosynthesis-related gene cluster signatures in the microbiomes of Sof Umer Cave have not been explored. Therefore, high-throughput shotgun sequencing was used to explore biosynthesis-related gene clusters (BGCs) in the microbiomes of Sof Umer Cave. The GeneAll DNA Soil Mini Kit was used to extract high-molecular-weight DNA from homogenized samples, and the purified DNA was sequenced using a NovaSeq PE150. According to the Micro-RN database, the most common microbial genera in Sof Umer Cave are *Protobacteria, Actinobacteria, Verrucomicrobiota,* and *Cyanobacteria*. The biosynthesis-related gene clusters were annotated and classified, and the BGCs were predicted using antiSMASH and NAPDOS1. A total of 460 putative regions of BGCs encoding a wide range of secondary metabolites were identified, including RiPP (47.82%), terpene (19.57%), NRPS (13.04%), hybrid (2.18%), and other newly annotated (10.87%) compounds. Additionally, the NAPDOS pipeline identified a calcium-dependent antibiotic gene cluster from *Streptomyces coelicolor*, an actinomycin gene cluster from *Streptomyces chrysomallus*, and a bleomycin gene cluster from *Streptomyces verticillus*. These findings highlight the untapped biosynthetic potential of the Sof Umer Cave microbiome, as well as its potential for the discovery of natural products.

## 1. Introduction

In recent years, there has been a surge of interest in investigating microbial communities in various environments to discover new bioactive substances that could be used in medicine, agriculture, and industry [1,2]. Caves are ecosystems that harbor commercially important microbes that are able to produce secondary metabolites [3]. However, cave ecosystems have

Information was submitted with the manuscript.

**Funding:** The author(s) received no specific funding for this work.

**Competing interests:** The authors declare that they have no competing interests.

not yet received attention, and unexplored and distinct environments exist for exploring unique microbial communities and novel genetic resources [4]. Various parameters, such as the absence of sunlight and limited organic matter, make a unique ecosystem in which microorganisms rely on alternative sources of energy and nutrient recycling mechanisms [5]. Despite these challenges, cave ecosystems thrive, supporting a wide range of microbial species that have adapted to underground environments [6]. These microorganisms have developed specific metabolic pathways to survive in harsh conditions, making caves promising sources for discovering new biosynthesis-related gene clusters (BGCs) [7].

Several studies have identified microbial communities and biosynthesis-related gene clusters from cave microbiomes. For example, Samanta et al. [8] reported various ranges of microbial community structure, metabolic potential, and biosynthesis-related gene cluster diversity from subterranean caves in India. In addition, Wiseschart et al. [9] explored the microbial community structure and its metabolic potential in Manao-Pee Cave, Thailand. Similarly, Cheng et al. [10] confirmed the microbial community composition and co-occurrence network of methano-trophs and bacteria in subterranean karst caves. Sof Umer Cave is an exceptional cave ecosystem renowned for its vast chambers, underground streams, and diverse microorganisms [4]. The exceptional isolation and favorable conditions of Sof Umer Cave make it an ideal location for studying microbial diversity and potential biosynthesis-related gene clusters or genetic resources.

Microbial genomes contain gene clusters related to biosynthesis (BGCs), which encode enzymes responsible for producing secondary metabolites [11]. BGCs are groups of genes that are localized together and are responsible for the synthesis of secondary metabolites. Secondary metabolites include nonribosomal peptide synthases (NRPSs), polyketide synthases (PKSs), ribosomally synthesized and posttranslationally modified peptides (RiPPs), and ter-penes, which are organic molecules produced by microorganisms [12]. These compounds play crucial roles in the ecology of microorganisms, functioning as defense mechanisms, signaling molecules, and means of communication [13].

Tools such as the antiSMASH pipeline [14] and NAPDOS pipeline [15] provide valuable support for the in silico detection and analysis of various types of BGCs, greatly advancing our understanding of ecologically and evolutionarily significant environments [16]. However, there is a notable gap in our knowledge of BGCs in microbial caves, which are complex eco-systems that host diverse microbial communities [17]. Microbes from caves, with their ancient origins and ability to withstand extreme environmental conditions, present a unique opportu-nity to explore the diversity and ecological functions of BGCs [16].

Despite the extensive research conducted on microbial caves, few studies have focused on investigating the presence and potential importance of BGCs in cave ecosystems. Therefore, this study aimed to fill this gap by using a high-throughput shotgun sequencing approach to explore the diversity of biosynthetic signatures of cluster genes (BGCs) in the microbiome of Sof Umer Cave and elucidate the structure of secondary metabolites. This objective was achieved through comprehensive analyses of metagenome-assembled genomes using the antiSMASH and NAPDOS pipelines, providing insights into the biotechnological potential of the microbiome of Sof Umer Cave and its secondary metabolites. Finally, the analysis revealed the diverse range of BGCs, highlighting the promise of the microbiome of Sof Umer Cave in the discovery of natural products and leading to drug development.

## 2. Materials and methods

### 2.1. Study area and sample collection

Sof Umer Cave, located in the Oromia region, Bale district Zone at an altitude of 1,269 m (6°51′0″-6°54′0″N latitude and 40°51′0″E longitude), is one of the largest caves in East Africa,

approximately 15.1 km in length. Inside, the cave's height reaches 60 m, whereas its width expands to 100 m in certain areas. The temperatures within the cave are relatively stable, ranging from a minimum of 19 to 21 °C to a maximum of 33 to 35 °C throughout the year. The surrounding soil has a pH range of 8.00–8.30 and is characterized by a sandy texture and sedimentary rock [5].

The permission was obtained from Bale district administration to collect samples from Sof Umer Cave. The samples were collected from walls, floors, and sediment deposits, were subsequently collected from various locations within Sof Umer Cave. A total of 1800 g of sample was collected from 600–1000 m from the cave entrance, as well as from the ground, dark zone, and surface of the cave (S1 Fig). The collected samples were homogenized, placed in sterile polyethylene bags, transported in an icebox, and stored at 4°C until laboratory analysis. Overall, the results of the metagenomic DNA extraction, sequencing, and sequence analysis are presented in the following flowcharts in Figs 1 and 2, respectively

## 2.2. Extraction of Environmental DNA

Environmental DNA (eDNA) was extracted from a composite sedimentary rock sample using the GeneAll DNA Soil Mini Kit (GeneAll Biotechnology, Co., Ltd.). 05729, Seoul, Korea).

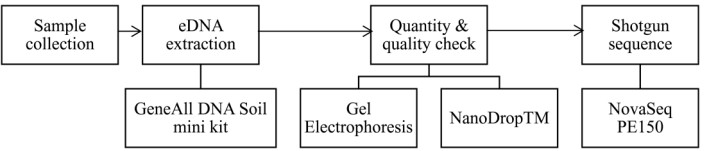

**Fig 1. Workflow of metagenomic DNA extraction and sequencing.**

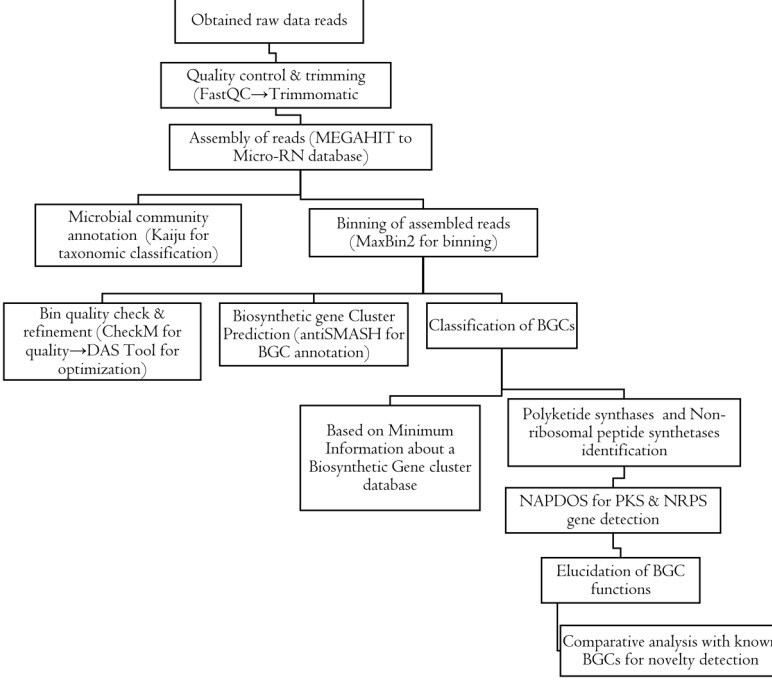

**Fig 2. Workflow for raw data analysis and biosynthetic gene cluster mining in Sof Umer Cave microbiomes.**

The integrity and quantity of the extracted metagenomic DNA were checked by gel electrophoresis (S2 Fig), whereas the quality was checked using a Thermo Scientific NanoDrop™ 3300 Fluorospectrometer (Wilmington, Delaware, USA) (S1 Table). DNA samples with high nucleic acid content (ng/µl) and standard absorbance (260/280) were pooled before shotgun sequencing.

### 2.3. Shotgun sequence

The metagenomics DNA was fragmented randomly into small pieces (>500 bp). The fragmented environmental DNA was subsequently repaired and ligated with Illumina adapters. The fragmented DNA was then selected on the basis of size, amplified through PCR, and purified using standard protocol. Purified metagenomics DNA (50 ng/µL) was used to generate a metagenomic library. The quality of the library was assessed using Thermo Fisher Qubit fluorometry and real-time PCR to determine the size distribution. After evaluated, the library was barcoded, pooled, and sequenced using shotgun sequencing in one lane of a flow cell using a NovaSeq PE150 (Illumina, Tsim Sha Tsui, Hong Kong) with a 150-bp paired-end run.

### 2.4. Quality control and preprocessing

The quality of the reads was assessed using FastQC (version 0.11.6) [18]. Low-quality reads, adapter sequences, and read trimming were removed using Trimmomatic (version 0.36) [19] (S3 Fig).

### 2.5. Metagenomic assembly and diversity analysis

High-quality reads were assembled into contigs using MEGAHIT (version 1.2.9) [20]. The assembled genes were then binned using MaxBin2 (version 1.1.1) [21], and the completeness and contamination levels were evaluated using CheckM (version 1.0.12) [25]. Five hundred base pair (bp) scaftigs were selected for open reading frame (ORF) prediction using GeneMark. Hmm (version 2.1) [22]. These predicted ORFs (S4 Fig) were dereplicated using CD-HIT (version 4.5.8) [23] to produce nonredundant gene catalogs, with a focus on contiguous gene sequences that produce nucleic acids. The clean data were then matched with the gene catalog using Bowtie2 (version 2.2.4) [24] to calculate gene abundance. Microbial diversity was determined by comparing the annotation of assembled metagenomic reads to the reference Micro-NR database in DIAMOND (version 2.1.6) [26] (S5A and S5B Fig).

### 2.6. Detection and annotation of biosynthesis-related gene clusters

The bins were generated by MaxBin2 version 1.1.1 [27] and optimized by DASTool to generate metagenome-assembled genomes (MAGs). Biosynthesis-related gene clusters (BGCs) were detected using the antiSMASH pipeline [14], which employs a comprehensive approach to identify regions containing these gene clusters. AntiSMASH considers sequence patterns, domain architecture, and enzyme functions to accurately detect BGCs.

### 2.7. Classification of biosynthesis-related gene clusters

BGCs are classified into various classes, including polyketide synthases (PKSs), nonribosomal peptide synthetases (NRPSs), ribosomally synthesized and posttranslationally modified peptides (RiPPs), and terpene synthases and hybrids. This classification is performed via databases such as MIBiG (Minimum Information about a Biosynthesis-related Gene Cluster) and Pfam. It enables the prediction of potential chemical scaffolds and biosynthetic pathways encoded by BGCs [14].

## 2.8.  Elucidation of NRPS and PKS clusters

To elucidate the chemical structures and possible biological functions encoded by known biosynthesis-related gene clusters, secondary metabolites were predicted using NAPDOS [15]. The NAPDOS web server was used with default settings for the C and KS domains, and modeling was performed using prediction algorithms on the metagenome-assembled genomes (MAGs).

## 3.  Results and discussion

### 3.1.  Summary of Sof Umer Cave metagenomic data

A total of 94,834,808 raw data reads were obtained, with an average length of 969 bp. This number included reads related to adapters (278,048 or 0.29%), nonreading reads (2,218 or 0.00%), and low-quality reads (0.00%). After completing quality control, 94,554,542 sequence reads (99.70%) were found to be suitable for bioinformatics analysis (Tables 1 and 2).

During the genome investigation, 218,891 open reading frames (ORFs) were identified. Analyzing the integrity of these ORFs generated interesting results. Overall, approximately 31.26% (68,428 ORFs) showed integrity. On the other hand, approximately 23.18% (50,728 ORFs) were found to be intact. Interestingly, 17.78% (38,924 ORFs) lacked integrity at both the start and end. However, a significant proportion of the ORFs (27.78% or 60,811) remained intact throughout. This comprehensive investigation was conducted on a genome with a total length of 118.29 Megabase pairs (Mbp), resulting in an average ORF length of 540.4 bp (S2 Table and S4 Fig).

### 3.2.  Microbial composition of Sof Umer Cave

The microbial composition of Sof Umer Cave was determined by Kraken [28] and the Micro-RN database. Among the total reads, 98% were categorized into distinct kingdoms (S5A Fig), with the bacterial domain accounting for 80% of the total reads. The analysis revealed that bacteria were the dominant group in the sedimentary rocks of Sof Umer Cave, accounting for 96% of the microbial community. Shotgun sequence analysis confirmed that distinct bacterial phyla were identified in the sedimentary rocks of Sof, Umer Cave (S5B Fig).

The dominance of bacteria in Sof Umer cave environments reflects their exceptional adaptability and metabolic versatility. This study aligns with the study reported by Zada et al. [4], which confirmed the high abundance of microbial diversity in two geochemically and mineralogically different caves. The diversity of microbes plays crucial roles in ecological processes such as organic matter degradation, nitrogen cycling, and the biosynthesis of secondary metabolites [8,11,29]. Bacterial dominance underscores the importance of bacteria in sustaining cave ecosystems and driving key biochemical processes

**Table 1.  The sequence quality metrics of the Sof Umer Cave rock sample.**

| Sample ID | Total raw data reads | Clean read (%) | Error (%) | Q20 (%) | Q30 (%) | GC (%) |
|---|---|---|---|---|---|---|
| SCR01 | 94,834,808 | 99.70%) | 0.03 | 97.36 | 93.48 | 66.31 |

**Table 2.  Statistics for scaftigs (≥500 bp).**

| Sample ID | Total length (bp) | Scaftigs number | Average length (bp) | N50 length (bp) | N90 length (bp) | Max length (bp) |
|---|---|---|---|---|---|---|
| SCR01 | 138,701,566 | 143,139 | 969.00 | 898 | 538 | 344,381 |

In addition, shotgun sequencing revealed economically important bacterial genera and species in Sof Umer Cave. Notably, the dominance of *Stenotrophomonas, Actinomadura* and *Streptomyces*, which are known for their ability to produce antibiotics, highlights the ecological and pharmaceutical importance of Sof Umer Cave, which aligns with studies reported by Samanta et al. (8). Other genera, such as *Pseudonocardia, Actinoallomurus, Amycolatopsis, Solirubrobacter*, and *Nocardioides,* contribute to the complexity of the ecosystem, while the presence of unassigned taxa suggests the possibility of potential novel species with undiscovered functions (Fig 3A and 3B).

In addition, *S. indicatrix, S. maltophilia,* and *S. lactitubi* were the bacterial species that dominated Sof Umer Cave. These species are known for their biosynthesis and nutrient cycling, which is supported by Wang et al. [30]. Additionally, *Carbonactinospora thermoautotrophica Chloroflexota, Verrucomicrobiota* and other unidentified bacteria suggest adaptations to unique nutrient sources, such as chemosynthesis [31,32]. The presence of the fungus Rhizopus arrhizus introduces a eukaryotic dimension that may aid in organic matter decomposition and nutrient cycling. The identification of unassigned taxa highlights the need for further research, potentially revealing novel microbial interactions and functions. Future studies should focus on characterizing these microbial communities and their responses to environmental changes, which will be essential for predicting the resilience of cave ecosystems.

### 3.3. Identification and classification of biosynthesis-related gene clusters

A total of 460 putative biosynthesis-related gene clusters (BGCs) were characterized, indicating a wide range of pathways for secondary metabolite biosynthesis in the microbiomes of Sof Umer Cave (S6 Fig and S3 Table). Most of these BGCs were associated with known types of secondary metabolites. Among them, the most abundant were RiPPs, accounting for 47.82% of the total. RiPPs encompass a diverse array of bioactive compounds, including lanthipeptides and LAPs, which are well known for their antimicrobial, anticancer, and immunomodulatory effects [33]. Terpenes represented a significant portion (19.57%) of the identified BGCs, highlighting their importance in microbial ecology and biotechnology due to their various biological activities, such as antimicrobial, antioxidant, and anticancer effects, making them

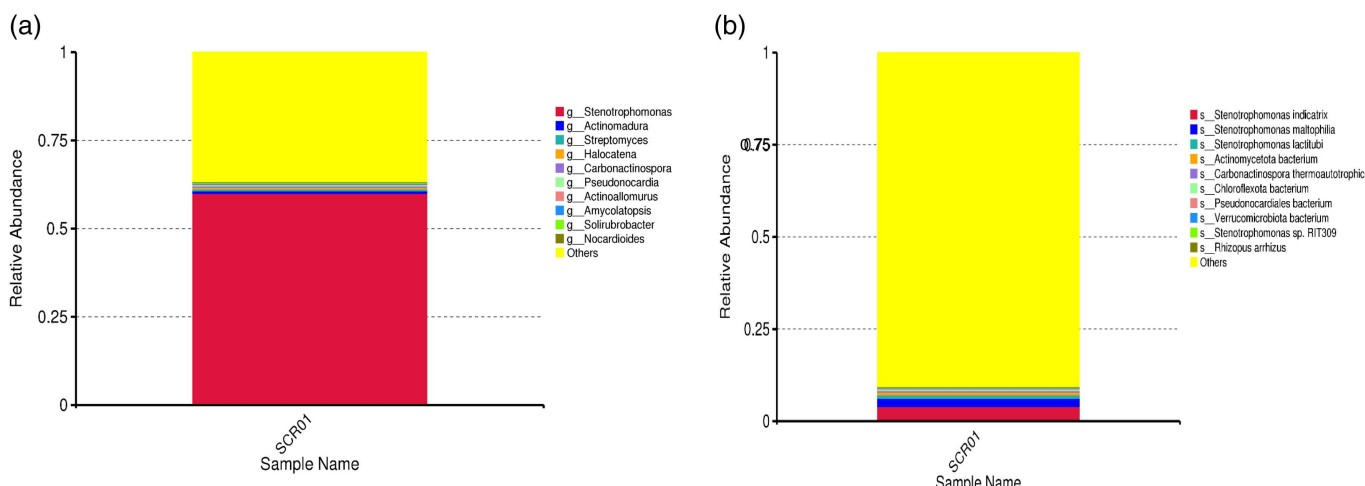

**Fig 3. (A) The genus-level taxonomic abundance in Sof Umer Cave, analyzed using Kaiju program. (B)** The species-level taxonomic abundance in Sof Umer Cave, analyzed using Kaiju program.

promising candidates for drug development and industrial applications, which aligns with studies reported by Samanta et al. [8] and Yang et al. [29].

The identified BGCs also included 13.04% NRPSs and 6.53% Type III polyketide synthases (T3PKS) (Fig 4). NRPSs and T3PKSs are versatile enzymatic assembly lines that play vital roles in producing structurally complex peptides and polyketides with medicinal and agricultural applications.

A small percentage (2.18%) of the detected BGCs were hybrid clusters, which combine features from distinct biosynthetic pathways. These hybrid clusters demonstrate the flexibility of microbial metabolism [34,35], enabling the production of novel bioactive compounds through combinatorial biosynthesis. Additionally, 10.87% of the detected BGCs were either newly annotated or unrecognized by antiMASH analysis, suggesting potential novel biosynthetic pathways whose roles and biological activities require further research.

### 3.4. Cluster-blast comparative gene analysis

The data obtained from antiSMASH for the selected biosynthesis-related gene cluster (BGC) at k127_25565 (Region 27-PKS) revealed several genomic regions from *Stenotrophomonas* sp. with the highest percentages of similarity via cluster BLAST. These regions were used for comparative gene clustering and alignment of gene clusters homologous to the query cluster analysis (Fig 5). The alignment focuses on the top ten hits related to the calcium-dependent antibiotic KPS gene cluster of *Stenotrophomonas indicatrix* strain EYE 224. The homologous genes, identified through BLAST with a maximum sequence identity of 100% and shortest BLAST alignment at 100% of the sequence, are color-coded identically. These alignments include PubMed and/or PubChem links for gene clusters related to the *Stenotrophomonas* sp. YAU14D1 LEIMI41 strain, which shows 97% similarity.

Specifically, the *Stenotrophomonas indicatrix* strains EYE 224, *Stenotrophomonas sp.* DR822, and *Stenotrophomonas maltophilia* each presented 100% gene similarity to the reference sequences. These strains are associated with arylpolyene biosynthesis-related gene clusters, as indicated by the presence of the partial terms "aryl," "arylpol," and "arylp" in the data. Additionally, the *Stenotrophomonas maltophilia* strains MER1, *Stenotrophomonas* sp. ESTM1D MKCIP41, and LH-B2 presented a slightly lower similarity of 97% but still demonstrated strong genetic correlation. The consistent presence of arylpolyene-related genes across these strains underscores their potential for producing arylpolyene compounds, which are

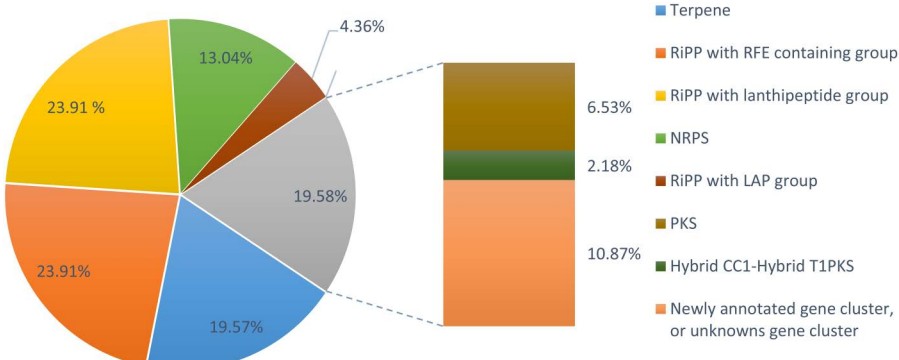

**Fig 4. Genomic regions annotated and predicted the distribution of various BGC types using antiSMASH.** RiPPs with an LAP group (23.91%), RiPPs with a lanthipeptide group (23.91%), terpenes (19.57%), NRPSs (13.04%), hybrid clusters (2.18%), and other newly annotated compounds (10.87%).

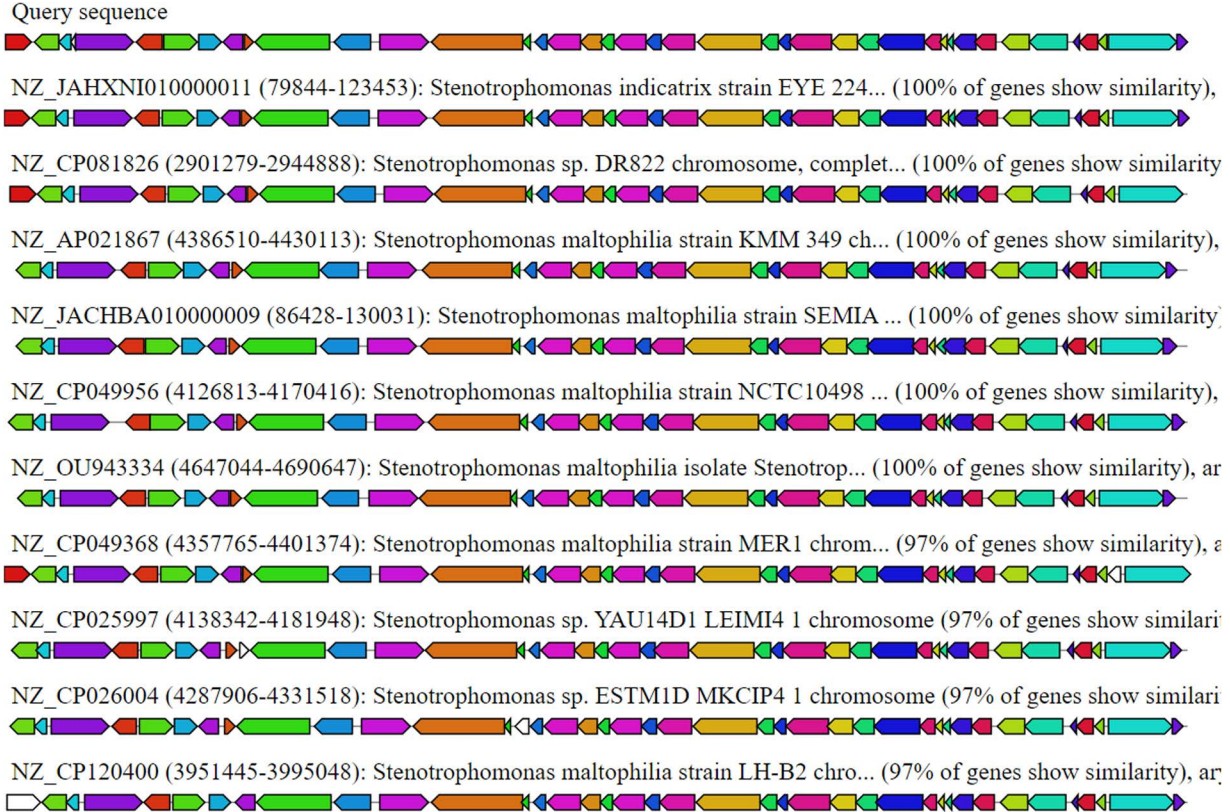

**Fig 5. Putative regions containing gene clusters associated with biosynthesis.** The alignment, conducted via Cluster-BLAST, revealed gene clusters that are related to the queried gene cluster. Specifically, the top ten matches to the setomimycin and kinamycin antibiotic KPS gene cluster from the Stenotrophomonas indicatrix strain EYE 224 are shown. Genes with significant homology (BLAST; highest sequence identity of 100%; BLAST alignment covering 100% of the sequence) are color-coded identically.

known for their diverse biological activities. This genomic coherence suggests that these *Stenotrophomonas* strains could be valuable for biotechnological applications involving arylpolyene biosynthesis.

These findings emphasize the role of genetic conservation and horizontal gene transfer in shaping microbial secondary metabolism and natural product diversity [9,36]. Furthermore, the discovery of conserved gene clusters among various bacterial strains highlights the potential for bioprospecting and biotechnological applications. Understanding the genetic basis of biosynthetic pathways enables researchers to manipulate and design microbial strains to produce novel bioactive compounds with medicinal, agricultural, and industrial applications [37, 38].

Another biosynthesis-related gene cluster (BGC), K127_195235-Region 09-posttranslationally modified peptides (RiPPs) (Fig 6), indicates strong potential for the production of RiPPs. Strains such as *Stenotrophomonas indicatrix* strain D763, *Stenotrophomonas maltophilia* strain U5, *Stenotrophomonas indicatrix* strain DAIF1, *Stenotrophomonas* sp. MYb57 and *Stenotrophomonas maltophilia* strain MER1 all show 100% gene similarity with known reference sequences, highlighting their genetic coherence. These strains were annotated with terms such as "RiPP," "RiP," "RiPP-like," and "RiPP-," indicating their association with RiPP biosynthetic pathways.

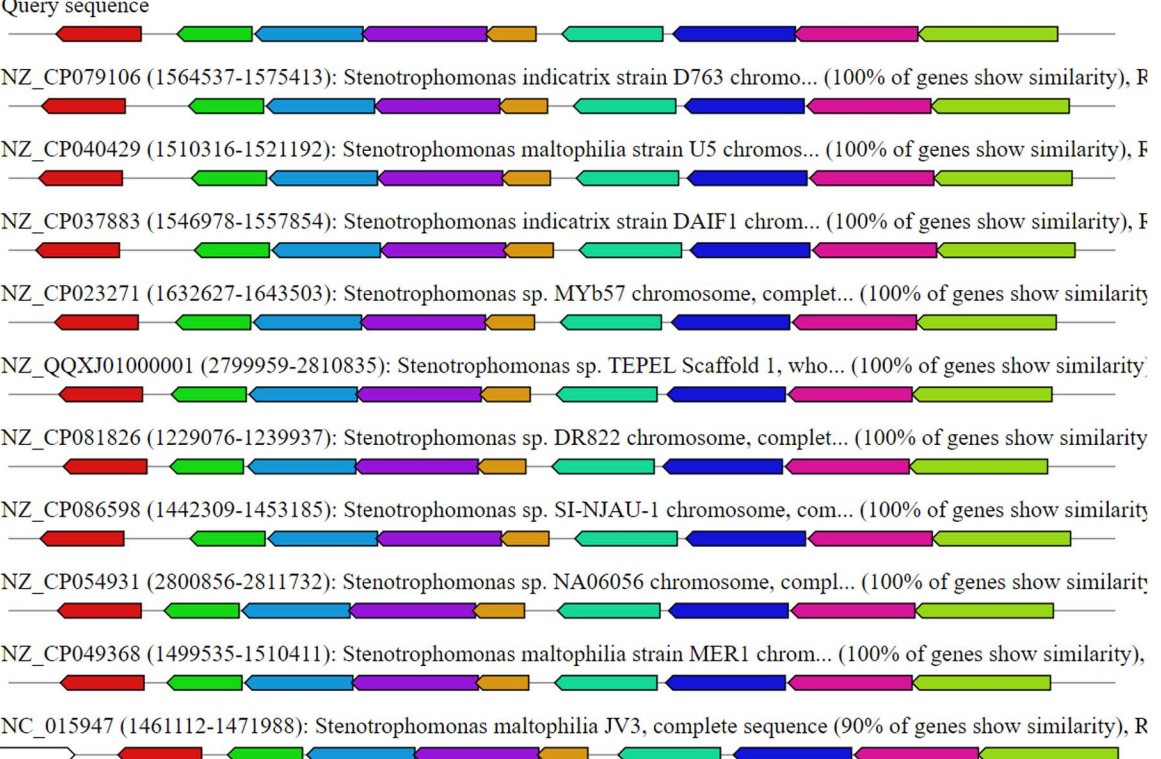

**Fig 6. Putative region contains the biosynthesis-related gene cluster, K127_195235-Region 09-posttranslationally modified peptides.** The alignment, which was conducted via Cluster-Blast, revealed the gene clusters with the highest similarity to the queried gene cluster.

The strain *Stenotrophomonas maltophilia* JV3 shows slightly lower genetic similarity at 90% but still maintains a significant association with the RiPP pathway. This consistent genetic pattern across multiple strains suggests a strong and inherent capacity for RiPP production within the genus *Stenotrophomonas*. The high gene similarity indicates that these strains may share functional biosynthetic pathways, contributing to their potential as sources for novel RiPP compounds. Furthermore, compared with other biosynthesis-related gene clusters, the putative areas at k127_57222, area 1, and ectoine revealed poor similarity (S7 Fig).

### 3.5. Comparative analysis of biosynthesis-related gene clusters

The analysis of distinct biosynthesis-related gene clusters (BGCs) from various organisms provides valuable insights into their similarity scores, compound types, and ecological potential. The similarity scores, which range from 0.64 to 0.86, indicate different degrees of resemblance to known reference BGCs. Higher scores, such as 0.85 for BGC0002424 and 0.86 for BGC0001925, suggest a closer evolutionary relationship and potentially similar biosynthetic mechanisms. This is particularly exemplified by saccharochelin A and B produced by Saccharothrix sp., which highlight conserved pathways responsible for these nonribosomal peptides (NRPs), which are known for their roles in microbial competition [39]. In contrast, lower similarity scores, such as 0.64 for BGC0001548, may indicate more divergent and novel biosynthetic pathways, as observed with RiPP citrulassin A from *Streptomyces albulus*, which merits further exploration (Table 3).

**Table 3. MIBiG comparison analysis of distinct protoclusters in relation to reference regions and their respective properties.**

| Reference | Similarity score | Type | Compound(s) | Organism |
|---|---|---|---|---|
| BGC0002424 | 0.85 | NRP | Saccharochelin A & B | *Saccharothrix sp.* |
| BGC0002052 | 0.76 | Other (Ectoine) | Ectoine | *Streptomyces sp.* |
| BGC0001548 | 0.64 | RiPP | Citrulassin A | *Streptomyces albulus* |
| BGC0001925 | 0.86 | Alkaloid | Altemicidin, SB-203207, SB-203208 | *Streptomyces sp.* |
| BGC0002689 | 0.73 | NRP | 2,3-Dihydroxybenzoylserine | *Stenotrophomonas maltophilia* |
| BGC0000131 | 79 | Polyketide | Pyrrolomycin | *Streptomyces sp.* |
| BGC0001226 | 0.75 | RiPP | Streptocollin | *Streptomyces collinus* |
| BGC0000252 | 0.74 | Polyketide | Nonactin | *Streptomyces griseus* |
| BGC0002007 | 0.84 | Terpene | Atolypene | *Amycolatopsis tolypomycina* |

The identified compounds encompass a diverse array, including NRPs, polyketides, ribosomally synthesized and posttranslationally modified peptides (RiPPs), alkaloids, and terpenes, underscoring their ecological and therapeutic importance. For example, alkaloids such as altemicidin from Streptomyces sp. exhibit antimicrobial and cytotoxic properties [40], whereas polyketides such as nonactin from *Streptomyces griseus* play crucial roles in drug discovery. This diversity reflects the adaptive strategies of these organisms in various environments, with terpenes such as atolypene from *Amycolatopsis tolypomycina* indicating the production of volatile compounds that influence ecological interactions [41]. The abundance of diverse secondary metabolites among genera such as *Streptomyces* highlights their evolutionary resilience and potential as biotechnological resources for novel therapeutic agents.

### 3.6. Analysis of NRPS/PKS C and KS domain predictions

The NRPS/PKS domains were elucidated using NAPDOS, as were the newly identified NRPS/PKS protein domains and functional/phylogenetic subgroups. PKS/NRPS gene names are annotated on the basis of the domains and domain subtypes they contain, such as 'hybrid

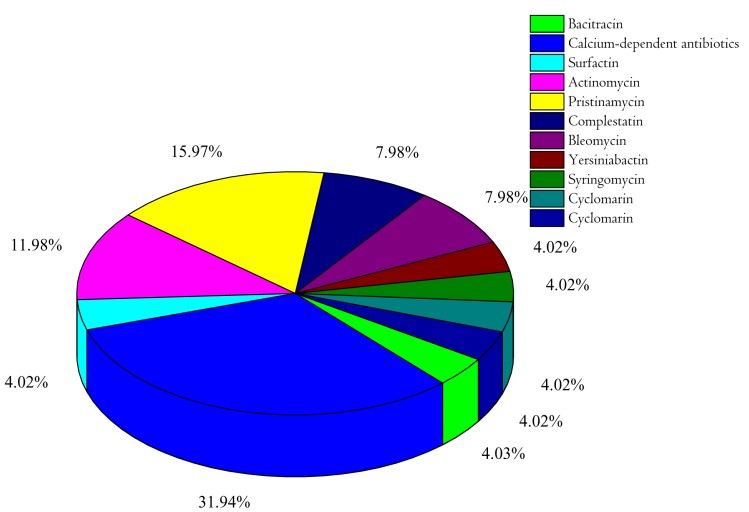

**Fig 7. Analysis of major secondary metabolites predicted by NAPDOS based on ketosynthase (KS) domains.** Yersiniabactin was identified as a siderophore (iron-chelating agent), syringomycin was identified as an antifungal agent, and the remaining were antibacterial agents.

NRPS-PKS', 'enediyne PKS', 'glycopeptide NRPS', and 'trans-AT PKS' (Fig 7 and S4 and S5 Tables). The secondary metabolites derived from various bacterial species in the Sof Umer Cave microbiome are generated by nonribosomal peptide synthetase (NRPS) or hybrid cluster types. For example, the bacitracin sequence from *B. licheniformis* showed 97% identity, with an alignment length of 88. Other routes of secondary metabolite production, such as the use of calcium-dependent antibiotics *from Streptomyces coelicolor*, actinomycin from *Streptomyces chrysomallus*, pristinamycin from *Streptomyces pristinaespiralis*, and bleomycin from *Streptomyces verticillus*, have been discovered.

The wide range of potential natural product biosynthetic pathways, as well as the variations in percent identity and alignment length between matches, reveal the genetic diversity of these pathways. This diversity suggests that microbial communities have broad functional potential for producing secondary metabolites with various chemical structures and biological functions [42–44]. Understanding the complex interactions between gene clusters and their related pathways is crucial for determining the secondary metabolite production capacity of microbial populations.

### 3.7. Chemical diversity and opportunities for the discovery of drugs from Sof-Umer Cave microbiomes

Predictive analysis of biosynthesis-related gene clusters (BGCs) revealed that the microbial community of the Sof Umer Cave microbiota harbored a variety of structurally diverse secondary metabolites [45]. These include compounds with antibacterial properties, antifungal agents, cytotoxic chemicals, and pigments, indicating the potential for identifying bioactive compounds for medicinal [9], agricultural [31], and industrial [46] purposes.

Our analysis of BGCs revealed the presence of a wide range of secondary metabolites in the Sof Umer Cave microbiome. These metabolites include various bioactive substances, such as antimicrobial drugs that can kill pathogenic microbes [6], antifungal compounds that target fungal pathogens [42,47], cytotoxic molecules with anticancer properties, and pigments with industrial applications [43].

The discovery of antimicrobial compounds is particularly important given the urgent need for new medicines to treat drug-resistant infections. Similarly, antifungal compounds offer promise in addressing agricultural issues caused by fungal diseases, while cytotoxic molecules could be utilized for the development of anticancer drugs. Furthermore, pigments produced by microbes from caves may have applications in various industries, such as food, textiles, and cosmetics [48].

The presence of terpenes, RiPPs (ribosomally synthesized and posttranslationally modified peptides), PKS (polyketide synthase), NRPS (nonribosomal peptide synthetase) (Fig 4), hybrids, and biosynthesis-related gene clusters (BGCs) greatly enhances opportunities for drug discovery and development. By thoroughly exploring and exploiting these natural product reservoirs, efforts in drug discovery can maximize the benefits of the vast variety and medicinal value present in these microbial metabolic pathways [49].

## 4. Conclusion

High-throughput shotgun sequencing of the Sof Umer Cave microbiome revealed many biosynthesis-related gene clusters (BGCs) that encode various secondary metabolites. The most abundant microbial genera found were *Protobacteria, Actinobacteria, Verrucomicrobiota,* and *Cyanobacteria.* The presence of 460 potential BGCs indicates a high capacity for producing secondary metabolites such as RiPP, terpenes, NRPS, and other chemical compounds. Notably, the identification of gene clusters encoding known antibiotics, such as *Streptomyces coelicolor, Streptomyces chrysomallus,* and *Streptomyces*

*verticillus,* highlights that Sof Umer Cave has potential as a resource of novel biologically active substances. This study confirms the presence of various biosynthesis-related gene clusters (BGCs) associated with the Sof Umer Cave microbiome and underscores their importance in natural product discovery. Future research and bioprospecting initiatives in these unique environments offer promising opportunities for drug development and biotechnology applications.

## Supporting information

**S1 Fig. (A) Samples taken from the wall of Sof Umer Cave. (B) Collected and homogenized samples.**
(DOCX)

**S2 Fig. Agarose gel electrophoresis of metagenomic DNA extracted from rocks from Umer Cave.**
(DOCX)

**S3 Fig. Output of Trimmomatic, a widely used tool for preprocessing high-throughput sequencing data.**
(DOCX)

**S4 Fig. The scaftig (scaffold) length distribution (scaftigs >500 bp) was used for open reading frame (ORF) prediction.**
(DOCX)

**S5A Fig. Taxonomy and distribution of microbial genomes at the bacterial level via Krona and the Micro-NR database.**
(DOCX)

**S5B Fig. Taxonomies and distribution of microbial genomes in the Actinobacteria phylum using Krona and the Micro-NR database.**
(DOCX)

**S6 Fig. Putative biosynthesis-related gene cluster regions annotated by antiMASH.**
(DOCX)

**S7 Fig. Selected putative regions of biosynthesis-related gene clusters annotated by antiMASH at 127_57222 - Region 1 - RRE-containing, lassopeptide.**
(DOCX)

**S1 Table. NanoDrop reads data of metagenomic DNA extracted from the rocks of Sof Umer Cave.**
(DOCX)

**S2 Table. Statistics of the gene catalogs.**
(DOCX)

**S3 Table. The secondary metabolite regions were identified via strictness 'relaxed' (truncated to the first 100 record(s).**
(DOCX)

**S4 Table. Biosynthesis-related gene clusters related to species.**
(DOCX)

**S5 Table. Major types of natural products.**
(DOCX)

## Author contributions

**Conceptualization:** Abu Feyisa Feyisa, Mesfin Tafesse Gemeda, Musin Kelel Abas, Gessesse Kebede Bekele.

**Data curation:** Abu Feyisa Feyisa, Mesfin Tafesse Gemeda, Musin Kelel Abas, Gessesse Kebede Bekele.

**Formal analysis:** Abu Feyisa Feyisa, Mesfin Tafesse Gemeda, Musin Kelel Abas, Gessesse Kebede Bekele.

**Investigation:** Abu Feyisa Feyisa, Mesfin Tafesse Gemeda, Musin Kelel Abas.

**Methodology:** Abu Feyisa Feyisa, Mesfin Tafesse Gemeda, Musin Kelel Abas, Gessesse Kebede Bekele.

**Project administration:** Abu Feyisa Feyisa.

**Resources:** Abu Feyisa Feyisa, Mesfin Tafesse Gemeda, Musin Kelel Abas, Gessesse Kebede Bekele.

**Software:** Abu Feyisa Feyisa, Mesfin Tafesse Gemeda, Gessesse Kebede Bekele.

**Supervision:** Mesfin Tafesse Gemeda, Musin Kelel Abas.

**Validation:** Abu Feyisa Feyisa, Mesfin Tafesse Gemeda, Musin Kelel Abas, Gessesse Kebede Bekele.

**Visualization:** Abu Feyisa Feyisa, Mesfin Tafesse Gemeda, Musin Kelel Abas, Gessesse Kebede Bekele.

**Writing – original draft:** Abu Feyisa Feyisa, Mesfin Tafesse Gemeda, Musin Kelel Abas, Gessesse Kebede Bekele.

**Writing – review & editing:** Abu Feyisa Feyisa, Mesfin Tafesse Gemeda, Musin Kelel Abas, Gessesse Kebede Bekele.

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
