## [Decision Letter · Decision Letter 0]

30 Sep 2024

PONE-D-24-37159
Exploring Bioactive Compound Origins: Profiling Gene Cluster Signatures Related to Biosynthesis in Microbiomes of Sof Umer Cave, Ethiopia
PLOS ONE

Dear Dr. Feyisa,

Thank you for submitting your manuscript to PLOS ONE. After careful consideration, we feel that it has merit but does not fully meet PLOS ONE’s publication criteria as it currently stands. Therefore, we invite you to submit a revised version of the manuscript that addresses the points raised during the review process.

We look forward to receiving your revised manuscript.

Kind regards,

Richa Salwan

Academic Editor

PLOS ONE

Journal Requirements:

https://doi.org/10.1007/s42452-024-06110-x

In your revision ensure you cite all your sources (including your own works), and quote or rephrase any duplicated text outside the methods section. Further consideration is dependent on these concerns being addressed.

“I have read the journal's policy, and the authors of this manuscript have the following competing interests: The authors declare that they have no competing interests.”

Reviewers' comments:

Reviewer's Responses to Questions

**Comments to the Author**

1. Is the manuscript technically sound, and do the data support the conclusions?

Reviewer #1: Yes

Reviewer #2: Yes

2. Has the statistical analysis been performed appropriately and rigorously? 

Reviewer #1: Yes

Reviewer #2: N/A

3. Have the authors made all data underlying the findings in their manuscript fully available?

Reviewer #1: Yes

Reviewer #2: No

4. Is the manuscript presented in an intelligible fashion and written in standard English?

Reviewer #1: Yes

Reviewer #2: Yes

5. Review Comments to the Author

Reviewer #1: Work done by author is good.

Title of the manuscript acceptable as work done by author

Abstract: all things mention clearly in the abstract. explained everything well

Introduction: well written

Material and methods: all the procedure used during work step wise clearly mentioned

Results and discussion: all the data clearly written in graphical manner with photographs. and discussed very well

Reviewer #2: Dear Authors,

The MS PONE-D-24-37159 entitled Exploring Bioactive Compound Origins: Profiling Gene Cluster Signatures Related to Biosynthesis in Microbiomes of Sof Umer Cave, Ethiopia can be accepted for publication after minor corrections.

What is eDNA?

In figure 2 Qualitaty correct type mistake. Revise figure there should be connectivity and flow of information in the form of steps

Line 119 amplified through PCR …. Mention complete details about the methodology how DNA was amplified?

Line 119 An effective concentration what it that mention in unit

At what place instrument used for quantification is nanaodrop whereas at other place it is qubit? There should be uniformity

Authors should data to public repository

6. PLOS authors have the option to publish the peer review history of their article (what does this mean?). If published, this will include your full peer review and any attached files.

Reviewer #1: No

Reviewer #2: **Yes: **Vivek Sharma

---

## [Author Response · Author response to Decision Letter 0]

4 Oct 2024

Dear Editors:

We would like to thank you for your invitation to resubmit the revised manuscript. We have addressed each point raised, and we believe that the revisions have improved the quality and clarity of our manuscript.

Response for Editors

1. Additional permits information in the Methods section.

o Response: Permission was not required for the collection of sedimentary rock samples. However, we announce the Bale district administration orally because our study was not applied on humans or animals.

2. Address overlapping text with previous publication (https://doi.org/10.1007/s42452-024-06110-x).

o Response: We have revised the manuscript to avoid overlap and appropriately cited any relevant material. The text has been rephrased, and the necessary citations have been included.

 Changes in the Manuscript: Specifically, some parts of the methods, results and discussion sections have been revised to avoid self-plagiarism. Please see the highlighted parts.

3. Competing interests.

o Response: We have submitted the competing interests online as per the journal's guidelines.

4. Captions for the Supporting Information.

o Response: We have added captions for all the Supporting Information files of the manuscript and updated the in-text citations accordingly.

 Change in Manuscript: Captions were added to the Supporting Information section.

5. Provide original uncropped and unadjusted images for blot or gel results.

6. Response: We have provided original, uncropped and adjusted images of the gel results.

Response to Reviewer 1

Dear Reviewer,

We would like to thank you for your valuable comments and suggestions. We have addressed each point raised, and we believe that the revisions have improved the quality and clarity of our manuscript. Below are our responses to each comment, as well as the changes made to the manuscript.

Reviewer #1:

1. Title of the manuscript is acceptable; abstract and introduction are well-written.

o Response: Thank you for your positive feedback.

2. Materials and methods: all the procedures are clearly mentioned.

o Response: We appreciate your comments, even though we have made slight modification to this section as per Reviewer 2.

3. Results and discussion: data is clearly written and well-discussed.

o Response: Thank you for your acknowledgment.

We believe these revisions address your concerns and significantly improve the quality of our manuscript. We are confident that the changes we have made have improve our findings and the clarity of our presentation. Please let us know if there are any other areas that require further attention.

Thank you again for your constructive feedback.

Response to Reviewer 2

Dear Reviewer,

We would like to thank you for your valuable comments and suggestions. We have addressed each point raised, and we believe that the revisions have improved the quality and clarity of our manuscript. Below are our responses to each comment, as well as the changes made to the manuscript.

Reviewer #2:

1. What is eDNA?

o Response: We have added a definition of eDNA in the methodology section to clarify this term for readers who may not be familiar with it.

 Change in Manuscript: “Environmental DNA (eDNA)” refers to DNA collected from environmental samples such as soil.

2. In Figure 2, "Qualitaty" should be corrected to "Quality."

o Response: We have corrected the type error in Figure 2 and ensured clarity and flow in the figure. The revised figure now shows a more logical progression of information, with steps clearly indicated.

 Changes in the Manuscript: Figure 2 has been updated.

3. Line 119: Amplified through PCR – please provide complete details of the methodology.

o Response: We have revised this section to include the complete details of the PCR process, including the reagents, thermal cycling conditions, and specific primers used.

 Change in Manuscript: "eDNA was amplified using PCR with standard PCR protocol"

4. "An effective concentration" – please mention in units.

o Response: We have revised the manuscript to mention the concentration in specific units.

 Change in Manuscript: "50 ng/μL purified environmental DNA was used for ……………………"

5. Uniformity between Nanodrop and Qubit measurements.

o Response: We have ensured consistency in referring to the quantification instrument and clarified where the Nanodrop and Qubit instruments were used.

6. Data should be deposited in a public repository.

Response: We have submitted all raw data to the NCBI Sequence Read Archive (SRA) under accession number PRJNA1082540, and a supplementary file was submitted with the manuscript. The data can be accessed at the following link: (https://www.ncbi.nlm.nih.gov/sra/PRJNA1082540).

We believe these revisions address your concerns and significantly improve the quality of our manuscript. We are confident that the changes we have made have improve our findings and the clarity of our presentation. Please let us know if there are any other areas that require further attention.

Thank you again for your constructive feedback.

---

## [Editor Report · Decision Letter 1]

27 Nov 2024

Exploring Bioactive Compound Origins: Profiling Gene Cluster Signatures Related to Biosynthesis in Microbiomes of Sof Umer Cave, Ethiopia

PONE-D-24-37159R1

Dear Dr. Abu Feyisa Feyisa

We’re pleased to inform you that your manuscript has been judged scientifically suitable for publication and will be formally accepted for publication once it meets all outstanding technical requirements.

Kind regards,

Richa Salwan

Academic Editor

PLOS ONE
---

## [Editor Report · Acceptance letter]

PONE-D-24-37159R1

PLOS ONE

Dear Dr. Feyisa,

I'm pleased to inform you that your manuscript has been deemed suitable for publication in PLOS ONE. Congratulations! Your manuscript is now being handed over to our production team.

Kind regards,

on behalf of

Dr. Richa Salwan

Academic Editor

PLOS ONE